# Halogen–sodium exchange enables efficient access to organosodium compounds

Sobi Asako [1,2✉], Ikko Takahashi[2], Hirotaka Nakajima[1], Laurean Ilies[2] & Kazuhiko Takai [1✉]

With sodium being the most abundant alkali metal on Earth, organosodium compounds are an attractive choice for sustainable chemical synthesis. However, organosodium compounds are rarely used—and are overshadowed by organolithium compounds—because of a lack of convenient and efficient preparation methods. Here we report a halogen–sodium exchange method to prepare a large variety of (hetero)aryl- and alkenylsodium compounds including tri- and tetrasodioarenes, many of them previously inaccessible by other methods. The key discovery is the use of a primary and bulky alkylsodium lacking β-hydrogens, which retards undesired reactions, such as Wurtz–Fittig coupling and β-hydrogen elimination, and enables efficient halogen–sodium exchange. The alkylsodium is readily prepared in situ from neopentyl chloride and an easy-to-handle sodium dispersion. We believe that the efficiency, generality, and convenience of the present method will contribute to the widespread use of organosodium in organic synthesis, ultimately contributing to the development of sustainable organic synthesis by rivalling the currently dominant organolithium reagents.

[1] Division of Applied Chemistry, Graduate School of Natural Science and Technology, Okayama University, Okayama, Japan. [2] RIKEN Center for Sustainable Resource Science, Saitama, Japan. ✉email: sobi.asako@riken.jp; ktakai@cc.okayama-u.ac.jp

Since its discovery in the early 20th century, organolithium chemistry has played a dominant role in organic synthesis[1–7]. Organolithiums are archetypal organometallic compounds that have been used extensively as reactants and reagents (nucleophiles, bases, or reductants) for preparing a diverse range of organic, organometallic, and inorganic compounds. Common methods for preparing aryllithiums and congeners are the following: deprotonation of arenes, two-electron reduction of aryl halides, and halogen–lithium exchange between aryl halides and alkyllithiums (Fig. 1a)[3–11]. The halogen–lithium exchange method has been extensively used in organic synthesis because it allows rapid preparation of a large variety of organolithium compounds from (hetero)aryl or alkenyl bromides or iodides, typically using butyllithium or *tert*-butyllithium under cryogenic conditions. Because of its versatility and general scope, this method has become the standard entry to organometallic compounds and found widespread applications in various fields.

From a sustainability point of view, there is a growing demand for alternatives to the less abundant and increasingly expensive lithium. Sodium is the most abundant alkali metal and hence an attractive candidate. However, in contrast to the widespread use of organolithium compounds enabled by the accumulated knowledge of their chemistry, organosodium compounds have met with limited success in synthetic organic chemistry, even though organosodium chemistry first emerged as early as 1840–1850s[1–3,12]. This situation may reflect several unpractical features of organosodium compounds, such as their low solubility due to the high ionic character of sodium, and poor mechanistic knowledge caused in part by the less-informative nature of $^{23}$Na NMR.

Early attempts at halogen–sodium exchange using alkylsodiums such as butylsodium and pentylsodium met with significant challenges[13–19]. For example, Gilman reported that 1-bromonaphthalene reacted with butylsodium prepared from dibutylmercury and metallic sodium to afford 1-naphthylsodium in 28% yield (Fig. 1b); although details were not reported, we

speculate that the low yield may be partly due to the formation of side products such as 1-butylnaphthalene and octane (vide infra), and the preparation method using organomercury.

We have been exploring the potential of organosodium for organic synthesis for a while[20,21], and have recently reported that arylsodiums can be conveniently prepared by two-electron reduction of aryl chlorides with an easy-to-handle and highly reactive fine dispersion (particle size smaller than 10 μm) of sodium in paraffin oil (sodium dispersion; SD)[22], and subsequently participate in Negishi, Suzuki–Miyaura, and direct cross-coupling reactions (Fig. 1c)[20]. However, this method could only be applied for the preparation of a narrow range of organosodium compounds. Here, we report that a much broader range of aryl-, heteroaryl-, and alkenylsodium compounds are now accessible by halogen–sodium exchange between the corresponding organic bromides or iodides, and neopentylsodium prepared in situ from neopentyl chloride and SD, typically at 0 °C (Fig. 1d). Alkenyl bromides afforded the corresponding alkenylsodiums with retention of stereochemistry. Moreover, this method allowed us to efficiently access tri- and tetrasodioaryl compounds. The resulting organosodiums could be directly reacted with electrophiles, or used as nucleophiles for Negishi, Suzuki–Miyaura, and direct cross-coupling.

## Results

### Preparation of aryl- and alkenylsodium by halogen–sodium exchange with alkylsodium.

As depicted in Fig. 1b, halogen–sodium exchange is known to proceed with low efficiency, presumably because of the formation of side products from undesired reactions such as the reactions of organosodium with the alkyl halide through the Wurtz–Fittig reaction[23]. We envisioned that in order to suppress these undesired side reactions and achieve efficient halogen–sodium exchange, an alkylsodium compound should: (1) bear a bulky substituent to kinetically prevent Wurtz–Fittig coupling, (2) lack a β-hydrogen to avoid premature decomposition[1–3,24], (3) be

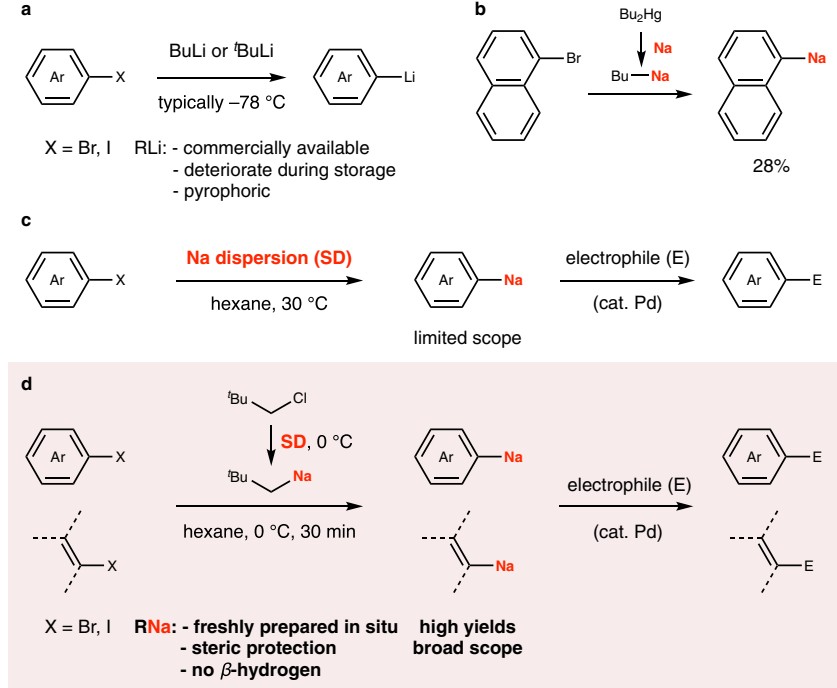

**Fig. 1 Preparation of organolithium and organosodium compounds. a** Halogen–lithium exchange. **b** Halogen–sodium exchange reported by Gilman[14]. **c** Our previous report: two-electron reduction of aryl halides with sodium dispersion[20]. **d** This report: halogen–sodium exchange with freshly prepared neopentylsodium.

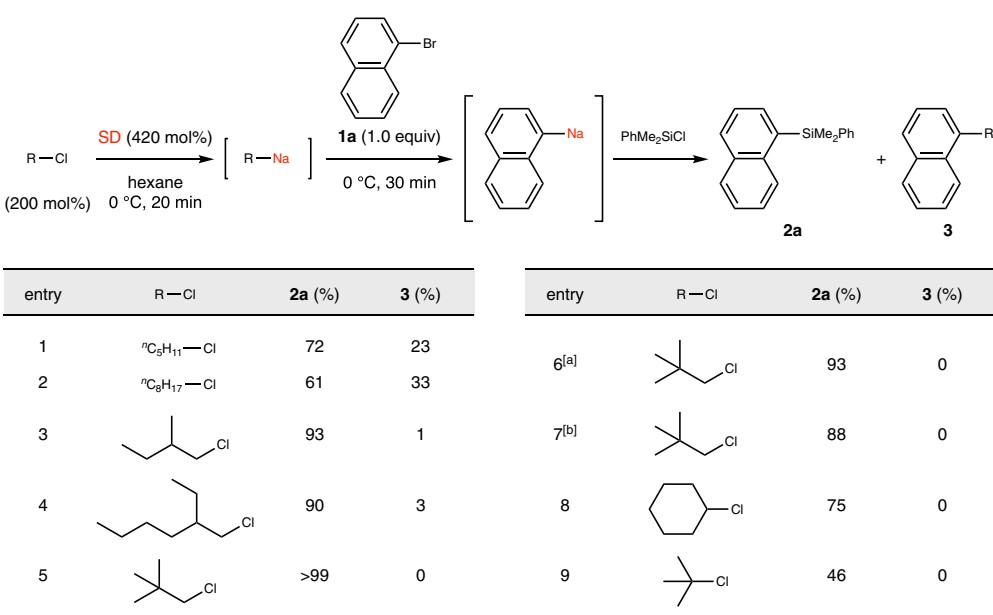

| entry | R—Cl | 2a (%) | 3 (%) |
|---|---|---|---|
| 1 | $^nC_5H_{11}$—Cl | 72 | 23 |
| 2 | $^nC_8H_{17}$—Cl | 61 | 33 |
| 3 | (3-methylbutyl structure)—Cl | 93 | 1 |
| 4 | (2-ethylhexyl structure)—Cl | 90 | 3 |
| 5 | (neopentyl structure)—Cl | >99 | 0 |

| entry | R—Cl | 2a (%) | 3 (%) |
|---|---|---|---|
| 6[a] | (neopentyl structure)—Cl | 93 | 0 |
| 7[b] | (neopentyl structure)—Cl | 88 | 0 |
| 8 | (cyclohexyl structure)—Cl | 75 | 0 |
| 9 | (tert-butyl structure)—Cl | 46 | 0 |

**Fig. 2 Bromine–sodium exchange between 1-bromonaphthalene and alkylsodium.** Alkylsodium prepared in situ from alkyl chloride (200 mol%) and sodium dispersion (SD: 420 mol%) was reacted with 1-bromonaphthalene (**1a**, 0.25 mmol) in hexane (2.0 mL) at 0 °C, followed by addition of PhMe$_2$SiCl. Yields were determined using [1]H NMR. [a]Neopentyl chloride (150 mol%) and SD (320 mol%). [b]Neopentyl chloride (120 mol%) and SD (250 mol%). *Ph* phenyl, *Me* methyl.

readily and quantitatively prepared in situ from a largely available alkyl halide, (4) be a primary alkylsodium, because it is known that secondary and tertiary alkylsodiums are generated sluggishly from the corresponding halides and undergo faster β-hydrogen elimination[1–3,24]. With the above considerations in mind, we commenced our study by identifying an appropriate alkylsodium for the bromine–sodium exchange using 1-bromonaphthalene (**1a**) as a model substrate (Fig. 2) to find that neopentylsodium[25] is an optimal reagent that maximizes the formation of 1-naphthylsodium, and minimizes the formation of undesired coupling product **3**. Thus, alkylsodiums were first prepared from the corresponding alkyl chlorides (200 mol%) and SD (particle size <10 μm, 420 mol%) in hexane at 0 °C, then reacted with **1a** (100 mol%) and trapped with PhMe$_2$SiCl to evaluate the efficiency of the reactions. The use of finely dispersed sodium is essential for the efficient and rapid preparation of alkylsodiums. We also recommend avoiding prolonged exposure of the SD to air, as its quality gradually deteriorates. We typically subdivided the SD into several dried vials and stored it under argon at −20 °C. If stored properly without exposure to air, the quality of the SD remains high for several months. We selected this substrate because 1-naphthylsodium could not be directly obtained (<10% yield) by two-electron reduction of **1a** with SD by following the method shown in Fig. 1c[15,20], probably owing to the stabilization of the radical anion species generated by one-electron reduction of **1a**. Although the reactions using pentyl chloride and octyl chloride afforded the desired 1-silylnaphthalene (**2a**), these were accompanied by the coupling byproduct as expected (entries 1 and 2). We were delighted to find that primary alkyl chlorides with bulky neighboring substituents indeed suppressed the side reactions, and neopentyl chloride performed particularly well among others to afford **2a** selectively in good yield (entries 3–5). The amounts of neopentyl chloride and SD could be reduced down to 120 and 250 mol%, respectively, but with slightly lower efficiency (entries 6 and 7). These organosodium compounds are largely insoluble in hexane, and we noticed that efficient and vigorous stirring is important for

efficiency and reproducibility. With efficient stirring, neopentylsodium could be typically prepared within 20 min and was found to be stable and reactive for at least 1 h at 0 °C in hexane before the addition of aryl halides (Supplementary Fig. 1). The reactions with secondary and tertiary alkyl chlorides such as cyclohexyl chloride and *tert*-butyl chloride were less efficient, partly because of the inefficient generation of the corresponding alkylsodiums (entries 8 and 9).

We then explored the scope of this procedure to find that a variety of aryl-, heteroaryl-, and alkenylsodiums were conveniently and rapidly accessible under mild conditions, as probed by trapping with electrophiles such as chlorosilanes and D$_2$O (Fig. 3). In addition to 1-bromonapthalene (**1a**), 1-iodonaphthalene (**1a'**), 2-bromonaphthalene (**1b**), and larger polycyclic aromatic hydrocarbons such as 9-bromophenanthrene (**1c**) and 1-bromopyrene (**1d**) reacted smoothly with silyl chlorides to afford the silylated products. 1-Chloronaphthalene afforded **2a** in low yield (ca. 10%). Aryl bromides possessing OMe (**1e**, **1f**), Cl (**1g**), F (**1h**), and CF$_3$ (**1i**) groups are known to undergo side reactions when reacted with alkylmetals, caused by the deprotonation of the hydrogen adjacent to these functional groups[26]. By treating these substrates with neopentylsodium at a lower temperature (−40 °C), these side reactions could be suppressed and the corresponding aylsodiums were obtained in good yields. 4-Bromophenol (**1j**), 4-bromobenzyl alcohol (**1k**), and 5-bromoindole (**1l**) participated in the reaction without requiring protection of the acidic protons, using an additional amount of neopentylsodium for deprotonation. Heteroaryl bromides such as 2-bromopyridine (**1m**), 8-bromoquinoline (**1n**), and 2-bromothiophene (**1o**) reacted at −78 to 0 °C. When 1,4-dibromobenzene (**1p**) and 2,2'-dibromo-1,1'-biphenyl (**1q**) were used as substrates, twofold sodiation proceeded, and disilylated benzene and silafluorenes were obtained after quenching the resulting disodioarenes with PhMe$_2$SiCl, Me$_2$SiCl$_2$, and Ph$_2$SiCl$_2$, respectively. The last reaction could be performed on a gram scale and the silafluorene product (**2r**) was obtained in good yield. Alkenyl bromides (**1s**, **1t**) reacted smoothly under these conditions. The reactions of 1-styryl bromide were

**Fig. 3 Halogen–sodium exchange between aryl and alkenyl halides and neopentylsodium.** Neopentylsodium prepared in situ from neopentyl chloride (200 mol%) and SD (420 mol%) was reacted with organic halides (**1**, 0.25 mmol) in hexane (2.0 mL) at 0 °C, followed by addition of R3SiCl (1.2 equiv), R2SiCl2 (2.0 equiv), or D2O (1.0 mL) as an electrophile. Yields determined by isolation are shown unless otherwise noted. Deuterium content was determined by [1]H NMR. [a]Me3SiCl (1.2 equiv). [b](Dodecyl)Me2SiCl (1.2 equiv). [c]Exchange reaction at −40 °C for 10 min. [d]Ratio of monosilylated and disilylated products were determined by [1]H NMR. [e]Neopentyl chloride (300 mol%), SD (630 mol%), hexane (2.5 mL), and PhMe2SiCl (3.0 equiv); after reaction, the concentrated crude mixture was reacted with K2CO3 (5.0 equiv) in MeOH (3.0 mL) for 1h. [f]Exchange reaction at −78 °C. [g]THF (0.80 mL) was used as a co-solvent. [h]Neopentyl chloride (120 mol%) and SD (250 mol%). [i]Exchange reaction for 10 min. [j]Neopentyl chloride (300 mol%), SD (630 mol%), hexane (2.5 mL), and PhMe2SiCl (3.0 equiv). [k]Neopentyl chloride (340 mol%), SD (710 mol%), hexane (2.5 mL), and R2SiCl2 (2.0 equiv). [l]4.0 mmol scale. *SD* sodium dispersion, *[t]Bu* tert-butyl, *Ph* phenyl, *Me* methyl, *MeO* methoxy.

stereoretentive, and the *E*/*Z* ratio of the substrate (**1t**) and product (**2t**) was identical. It should be noted that most of the organosodium compounds in Fig. 3 are not accessible by the direct reduction of aryl halides with SD (Fig. 1c) or by deprotonative sodiation.

Multimetalated arenes are valuable intermediates for rapid creation of molecular complexity, for example, in the synthesis of conjugated molecules of interest for materials science. Despite the synthetic promise of such species, the potential of multimetalation remains underutilized, even in organolithium chemistry[3,16,17,27,28]. Double sodiation via deprotonation has been reported[3,14,16,17,29–41], but efficient triple or quadruple sodiation has not been known to date. Our halogen–sodium exchange method enabled the regioselective access to tri- and tetrasodiated compounds (Fig. 4). Thus, 1,3,5-tris(4-bromophenyl)benzene (**1u**, Fig. 4a) and tetrakis(4-bromophenyl)methane

(**1v**, Fig. 4b) were tri- and tetrasodiated in good yield using neopentylsodium at 0 °C for 30 min, as confirmed by trapping with chlorosilane to afford **2u** and **2v**.

**Application to Negishi and Suzuki–Miyaura cross-coupling.** Following our previous studies[20], we explored whether arylsodium compounds prepared by bromine–sodium exchange could also be used as nucleophilic reagents in the palladium-catalyzed cross-coupling. We found that the arylsodiums smoothly underwent Negishi and Suzuki–Miyaura cross-coupling reactions after being transmetalated to organozinc and organoboron compounds using ZnCl2•TMEDA (TMEDA: *N,N,N′,N′*-tetramethylethylenediamine) or MeOBpin, respectively, in the presence of Pd-PEPPSI-IPr as catalyst (Figs. 5 and 6)[42]. The reactions were performed in a one-pot sequence: preparation of

**Fig. 4 Multisodiation through bromine–sodium exchange. a** Triple sodiation of **1u**: neopentylsodium prepared in situ from neopentyl chloride (450 mol%) and SD (950 mol%) in hexane (3.0 mL) at 0 °C for 20 min was reacted with 1,3,5-tris(4-bromophenyl)benzene (**1u**, 0.25 mmol, 1.0 equiv) at 0 °C for 30 min, followed by addition of PhMe$_2$SiCl (4.5 equiv). **b** Quadruple sodiation of **1v**: neopentyl chloride (600 mol%), SD (1260 mol%), tetrakis(4-bromophenyl)methane (**1v**, 0.20 mmol, 1.0 equiv), and PhMe$_2$SiCl (6.0 equiv). Yields determined by isolation are shown. SD sodium dispersion, $^t$Bu tert-butyl, Ph phenyl, Me methyl.

**Fig. 5 Palladium-catalyzed Negishi cross-coupling reactions using arylsodiums.** Conditions: step 1, neopentyl chloride (220 mol%) and SD (450 mol%) in hexane (2.0 mL) at 0 °C, 20 min; step 2, Ar$^1$Br (1.2 equiv) was added at 0 °C, 30 min; step 3, ZnCl$_2$•TMEDA (1.8 equiv) was added at 0 °C, stirred at RT for 30 min; step 4, Pd-PEPPSI-IPr (1 mol%), Ar$^2$Cl (0.30 mmol, 1.0 equiv), THF (0.80 mL), and N-methylpyrrolidone (NMP, 0.40 mL) were added and the cross-coupling reaction was performed at 70 °C for 3 h. Yields determined by isolation are shown. $^a$ZnCl$_2$•TMEDA (2.2 equiv). SD sodium dispersion, Ar aryl, $^t$Bu tert-butyl, TMEDA N,N,N',N'-tetramethylethylenediamine, THF tetrahydrofuran, NMP N-methyl-2-pyrrolidone, $^i$Pr isopropyl, Me methyl, Ph phenyl.

neopentylsodium, preparation of Ar$^1$Na via bromine–sodium exchange, transmetalation to Ar$^1$Zn or Ar$^1$B, and palladium-catalyzed cross-coupling with Ar$^2$Cl, to afford the coupling products in good to excellent yields. For example, naphthylsodium (i.e., **4a**, **5a**), arylsodiums bearing a diphenylamino (**4b**), carbazolyl (**5c**), trifluoromethyl (**4e**, **4f**), fluoro (**5b**), and tert-butyl (**4c**) substituent, heteroarylsodiums such as (methyl)thienylsodium (**4d**, **5f**) and quinolylsodium (**5d**), and styrylsodium (**5e**) were prepared in situ and found to function as efficient nucleophilic sources. As expected from the well-recognized functional group compatibility of these cross-coupling reactions, various functional groups on the electrophilic coupling partners such as

methoxycarbonyl (i.e., **4a**), carbonyl (**4c**), formyl (**5c**), cyano (**5a**), amino (**5d**), nitro (**5e**), trifluoromethyl (**4b**), trifluoromethoxy (**5d**), and methoxy (**5b**, **5e**) groups were tolerated. Overall, the halogen–sodium exchange we have developed here greatly expands the potential of organosodium-based cross-coupling technology.

**Application to direct cross-coupling**. The direct use of organosodium compounds in cross-coupling reactions circumvents the need for additional transmetalation, and therefore is an attractive, atom-economical, and less-wasteful alternative. After a short optimization of the reaction conditions that we previously developed for

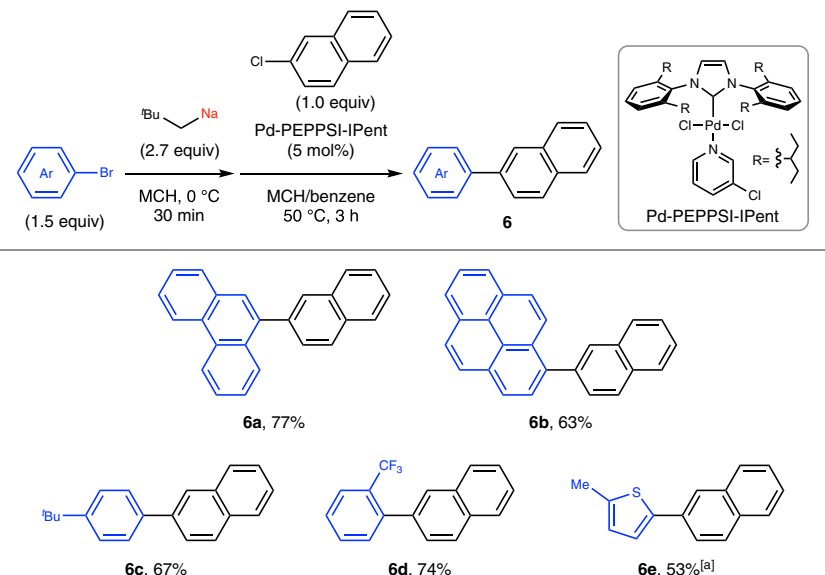

**Fig. 6 Palladium-catalyzed Suzuki–Miyaura cross-coupling reactions using aryl- and alkenylsodiums.** Conditions: step 1, neopentyl chloride (220 ml%) and SD (450 mol%) in hexane (2.0 mL) at 0 °C, 20 min; step 2, RBr (1.2 equiv) was added at 0 °C, 30 min; step 3, MeOBpin (1.2 equiv), and THF (0.80 mL) were added at 0 °C, 30 min; step 4, $H_2O$ (0.40 mL), Pd-PEPPSI-IPr (1 mol%), and ArCl (0.30 mmol, 1.0 equiv) were added and the cross-coupling reaction was performed at 70 °C for 5 h. Yields determined by isolation are shown. [a]Neopentyl chloride (250 mol%), SD (530 mol%), ArBr (1.4 equiv), and MeOBpin (1.4 equiv). [b]Pd-PEPPSI-IPr (2 mol%). [c]Neopentyl chloride (300 mol%), SD (630 mol%), hexane (2.5 mL), 2-bromothiophene (2.6 equiv), bromine–sodium exchange was performed for 10 min; MeOBpin (2.9 equiv), THF (1.0 mL), $H_2O$ (0.50 mL), 9,9-dioctyl-2,7-dibromofluorene (0.30 mmol, 1.0 equiv), and Pd-PEPPSI-IPr (2 mol%). SD sodium dispersion, [t]Bu tert-butyl, Ar aryl, MeOBpin 2-methoxy-4,4,5,5-tetramethyl-1,3,2-dioxaborolane; MeO methoxy; THF tetrahydrofuran, [i]Pr isopropyl.

**Fig. 7 Palladium-catalyzed direct cross-coupling reactions using arylsodiums.** Conditions: step 1, neopentyl chloride (270 mol%) and SD (570 mol%) in methylcyclohexane (MCH, 2.0 mL) at 0 °C for 20 min; step 2, ArBr (1.5 equiv) was added at 0 °C, 30 min; step 3, Pd-PEPPSI-IPent (5 mol%), 2-chloronaphthalene (0.30 mmol, 1.0 equiv), and benzene (1.0 mL) were added and the cross-coupling reaction was performed at 50 °C for 3 h. Yields determined by isolation are shown. [a]Coupling reaction for 1 h. SD sodium dispersion, Ar aryl, [t]Bu tert-butyl, Me methyl.

direct cross-coupling of organosodium compounds generated by two-electron reduction of aryl chlorides[20], the reactions of in situ-produced arylsodiums with 2-chloronaphthalene proceeded in good yields by using a Pd-PEPPSI-IPent catalyst in a solvent mixture of methylcyclohexane (MCH) and benzene (Fig. 7). It is noteworthy that these cross-coupling reactions did not require the slow addition of the organometallic reagent, as was the case with the direct coupling of organolithiums[43–45].

## Conclusion

Organosodium chemistry has long been overshadowed by well-established organolithium chemistry. Although there has been some recent renewed interest in the use of organosodium compounds for organic synthesis[46–58], the lack of general and reliable preparation methods has hindered the development of truly useful reactions. To change this status quo, we have demonstrated in this paper that efficient halogen–sodium exchange reactions

are now possible ~80 years after the seminal study[14], using neopentylsodium as a key metalating reagent, to expand the repertoire of available organosodium compounds greatly[59]. This exchange reaction has several attractive features: (1) the alkylsodium reagent can be conveniently and freshly prepared in situ using an inexpensive alkyl chloride and an easy-to-handle SD, circumventing the need for hazardous storage and transfer; (2) the SD in paraffin at 26 wt% concentration is easy-to-handle and less hazardous; this is a practical advantage, considering that organolithiums are highly pyrophoric and hazardous to use and store, and are the cause of many accidents in laboratories; (3) the reactions proceed typically at 0 °C using an ice bath; (4) sodium is ubiquitous, abundant, inexpensive, and therefore less vulnerable to supply risks; (5) multisodiated compounds, otherwise difficult to obtain, are reliably accessible. Thus, we believe that the reaction described here has the potential to replace the textbook halogen–lithium exchange, and open new frontiers for establishing organosodium-based organic chemistry.

## Methods

The SD in paraffin that we used in this study has an average particle size smaller than 10 μm and a concentration of ca. 26 wt%. It is non-pyrophoric and stable under air. Because of the lower reactivity of the SD at this concentration on contacting water, in Japan, SD is categorized as a less-hazardous material than commercial butyllithium. Although organosodiums generated in situ are potentially pyrophoric as are organolithiums, their concentrations in this study are kept low (0.2–0.4 M) and therefore their reactivity is mild.

**General procedure for the halogen–sodium exchange reactions**. In a dry Schlenk tube equipped with a glass-coated stirring bar, neopentyl chloride (0.50 mmol, 200 mol%) was added to a mixture of hexane (2.0 mL) and SD (26 wt%, 420 mol%) under nitrogen at 0 °C. After vigorously stirring at 0 °C for 20 min, an aryl halide (0.25 mmol, 1.0 equiv) was added at 0 °C and the reaction mixture was vigorously stirred for 30 min to form the corresponding arylsodium. The electrophile was added at the same temperature, and the reaction mixture was vigorously stirred at room temperature for 30 min. The reaction mixture was quenched with $H_2O$. After extraction with ethyl acetate three times, the combined organic layers were passed through a pad of silica gel and concentrated under reduced pressure. The crude product was purified by column chromatography on silica gel to afford the desired compound.

**General procedure for the Pd-catalyzed Negishi cross-coupling reactions using organosodium compounds prepared by bromine–sodium exchange**. In a dry Schlenk tube equipped with a glass-coated stirring bar, neopentyl chloride (0.65 mmol, 220 mol%) was added to a mixture of hexane (2.0 mL) and SD (26 wt%, 450 mol%) under nitrogen at 0 °C. After vigorously stirring at 0 °C for 20 min, an aryl bromide (0.36 mmol, 1.2 equiv) was added at 0 °C and the reaction mixture was vigorously stirred for 30 min to form the corresponding arylsodium. ZnCl₂•TMEDA (0.54 mmol, 1.8 equiv) was added at 0 °C and the reaction mixture was stirred at ambient temperature for 30 min to form the corresponding arylzinc. Tetrahydrofuran (THF, 0.80 mL), NMP (0.40 mL), Pd-PEPPSI-IPr (3.0 μmol, 1 mol%), and aryl chloride (0.30 mmol, 1.0 equiv) were sequentially added and the reaction mixture was stirred at 70 °C for 3 h. The reaction was quenched with a saturated aqueous solution of $NH_4Cl$. After extraction with ethyl acetate three times, the combined organic layers were passed through a pad of silica gel and concentrated under reduced pressure. The crude product was purified by column chromatography on silica gel to afford the desired compound.

**General procedure for the Pd-catalyzed Suzuki–Miyaura cross-coupling reactions using organosodium compounds prepared by bromine–sodium exchange**. In a dry Schlenk tube equipped with a glass-coated stirring bar, neopentyl chloride (0.65 mmol, 220 mol%) was added to a mixture of hexane (2.0 mL) and SD (26 wt%, 450 mol%) under nitrogen at 0 °C. After vigorously stirring at 0 °C for 20 min, an aryl bromide (0.36 mmol, 1.2 equiv) was added at 0 °C and the reaction mixture was vigorously stirred for 30 min to form the corresponding arylsodium. MeOBpin (0.36 mmol, 1.2 equiv) and THF (0.80 mL) were added at 0 °C and the reaction mixture was vigorously stirred for 30 min to form the corresponding arylboron compound, followed by the addition of $H_2O$ (0.40 mL) at the same temperature. Pd-PEPPSI-IPr (3.0 μmol, 1 mol%), and aryl chloride (0.30 mmol, 1.0 equiv) were sequentially added and the reaction mixture was stirred at 70 °C for 5 h. The reaction was quenched with a saturated aqueous solution of $NH_4Cl$. After extraction with ethyl acetate three times, the combined organic layers were passed through a pad of silica gel and concentrated under reduced pressure. The crude product was purified by column chromatography on silica gel to afford the desired compound.

**General procedure for the Pd-catalyzed direct cross-coupling reactions using organosodium compounds prepared by bromine–sodium exchange**. In a dry Schlenk tube equipped with a glass-coated stirring bar, neopentyl chloride (0.81 mmol, 270 mol%) was added to a mixture of MCH (2.0 mL) and SD (26 wt%, 570 mol%) under nitrogen at 0 °C. After vigorously stirring at 0 °C for 20 min, an aryl bromide (0.45 mmol, 1.5 equiv) was added at 0 °C and the reaction mixture was vigorously stirred for 30 min to form the corresponding arylsodium. Pd-PEPPSI-IPent (15 μmol, 5 mol%), 2-chloronaphthalene (0.30 mmol, 1.0 equiv), and benzene (1.0 mL) were sequentially added and the reaction mixture was stirred at 50 °C for 3 h. The reaction was quenched with a saturated aqueous solution of $NH_4Cl$. After extraction with ethyl acetate three times, the combined organic layers were passed through a pad of silica gel and concentrated under reduced pressure. The crude product was purified by column chromatography on silica gel to afford the desired compound.

## Data availability

The data that support the findings of this study are available from the corresponding author upon reasonable request. Detailed conditions for each reaction and compound characterization data are provided in the Supplementary Methods and Supplementary Figs. 1–5. NMR spectra are available in Supplementary Figs. 6–97.

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

## Acknowledgements

We thank Okayama University, RIKEN, and KOBELCO ECO-Solutions CO., Ltd. for financial support. We thank Dr. Zhaomin Hou and Dr. Masanori Takimoto for generously allowing us to use the mass spectrometer. This paper is dedicated to professor Dietmar Seyferth, who passed away on 6 June 2020.

## Author contributions

S.A., H.N., and K.T. started the project and later I.T. and L.I. joined. S.A., I.T., and H.N. conceived and designed the experiments. I.T. and H.N. performed the experiments. S.A., I.T., L.I., and K.T. co-wrote the manuscript. All authors contributed to discussions.

## Competing interests

S.A. and K.T. are listed as inventors on patent applications (PCT/JP2019/033980) that cover the halogen–sodium exchange reactions presented in this paper. All the other authors declare no competing interests.
