## [Peer Review File · Communications Chemistry]

Reviewers' comments:

Reviewer #1 (Remarks to the Author):

This work by Asako and Takai et al. describes concise generation of neopentylsodium and its use for halogen-sodium exchange. The resulting arylsodium species could be used for nucleophilic substitution with silyl chloride and for Suzuki/Negishi type Pd-catalyzed cross-coupling chemistry through transmetalation to B or Zn species as well as direct coupling with aryl chlorides, while the reaction conditions used for the cross coupling were almost optimized in the authors' previous works (Nat Catal 2019) and substantially the same protocols were adopted in these parts.

The key enabling the advance of the present work is use of neopentylsodium instead of conventionally used alkyl halides such as pentyl chloride, to avoid undesired side reactions and decomposition through beta-H elimination. I consider that the methodology presented here should be certainly useful as a new protocol for generation and use of neopentylsodium for halogen-sodium exchange. I thus recommend this work for publication in Communications Chemistry provided that the following point on scholarly presentation is addressed in the revision process.

In the introductory part, the authors criticized Gilman's pioneering work for aryl-sodium exchange, which was reported in 80 years ago. Indeed the reported yield of 1-naphthylsodium was lower (28% yield), while the butylsodium was prepared from dibutylmercury and metallic sodium. I considered that the problem of the Gilman's protocol may be on preparation of butylsodium from dibutylmercury, but not on the halogen-sodium exchange based on the fact that the authors examined use of normal-alkyl sodium for halogen-sodium exchange (Figure 2, entries 1 and 2), which gave 2a in reasonable yields along with over-substitution product 3 formed to some extent. As the Gilman's report did not clarify the details on the side products, I would urge the authors to revise this part by removing the authors' speculation.

Reviewer #2 (Remarks to the Author):

This is an extension of the authors previous work on arylsodium compounds that reported their preparation by reducing aryl chlorides with fine sodium dispersions and their application in Negishi, Suzuki–Miyaura, and cross-coupling reactions. The new improvement here is simply to use neopentylsodium to generate the arylsodiums in situ, and although this involves an extra step, it opens the door to a wider range of arylsodium compounds for onward reactivity via the same Pd catalytic and cross-coupling methods used previously. The scope of the halogen–sodium exchange methodology between aryl and alkenyl halides and neopentylsodium covered in figure 3 is impressive and is probably the best selling point of the work. This is original new chemistry which advances the use of organosodium compounds in situ in organic synthesis so merits publication.

(A) Specific Points on this paper:

(i) The paper states "organosodium reagents have limited success in synthetic organic chemistry". This requires elaboration to those chemists unfamiliar with the area. For example it should be explained that they have significant solubility problems; greater ionicity than RLi so generally are over reactive; and that ^{23}Na NMR is generally not very informative compared to the wealth of

information available from ^7Li and ^6Li NMR studies, which makes mechanistic interrogation of organosodium reactions much more challenging.

(ii) "The direct use of organosodium compounds in cross-coupling reactions circumvents the need for additional transmetalation". Progress in this objective would add to the interest and sustainability theme of the work. Unfortunately, this is limited to a single specialized case involving phenanthrenylsodium. Readers are left pondering, "what happens with the other sodium aryls mentioned in this paper?" Have they been tried?

(iii) On reflection if this was my work I would publish it as a full paper rather than a short communication. There are important aspects of the work which are relegated to the supporting information that should be more prominently displayed in the paper proper if this area is to be taken up by the organic synthetic community.

For example:

"Caution: organosodium compounds are insoluble in hexane, therefore efficient and vigorous stirring is very important".

"The quality of sodium dispersion gradually deteriorates upon contact with air."

"Once exposed to air, it is recommended that sodium dispersion should be used within two weeks".

"Neopentylsodium was stable at least for 1 h at 0 °C in hexane".

(iv) The chlorosilane used in the electrophilic quenches is always Me_2PhSiCl . What happens with Me_3SiCl ?

(v) The successful multi-sodiations of arenes is another impressive aspect of the study, which extends the potential impact of the work to material science. Other important strategies involving sodium could be cited here such as Martinez's work in *Science Advances* 2017, Vol. 3, no. 6, e1700832 "Multisodiation: Templated deprotonative metalation of polyaryl systems: Facile access to simple, previously inaccessible multi-iodoarenes".

(B) General points on influencing the thinking and take up by the field.

These are also important considerations of practicality that will influence whether organic chemists will embrace the use of such organosodium methods. How can the sodium dispersion be made in the laboratory? I cannot find these details in the manuscript or supporting information except that it is available from some commercial suppliers. This is critically important as the particle size of the sodium dispersion must be smaller than 10 μm and its concentration in paraffin must be controlled or else it could become very hazardous in terms of potential fires. The authors state "It is non-pyrophoric and stable under air. Because of the lower reactivity of the sodium dispersion at this concentration on contacting water, in Japan, the sodium dispersion is categorized as a less hazardous material than commercial butyllithium".

(i) For balance, it should be pointed out that organosodium compounds are generally more pyrophoric than organolithium compounds, and while the authors are discussing the sodium dispersion here they do not mention the potential pyrophoricity of the neopentylsodium reagent nor that of the alkenyl- and aryl-sodium intermediates they are producing following the initial dispersion step of the reaction. These comparisons should also be included in the paper.

(ii) Figure 1b states " RLi highly pyrophoric". Again for balance, should neopentylNa and the RNa not

also carry such warnings?

In summary, while the results in this study are important since they are building towards the aspiration of the development of an organosodium “technology”, the paper as written at present requires modification to give a more complete picture of the story so far focusing on important limitations as well as the promising advantages of the different types of organosodium compound that are involved.

Response to Reviewer 1 comments:

Comments:

This work by Asako and Takai et al. describes concise generation of neopentylsodium and its use for halogen-sodium exchange. The resulting arylsodium species could be used for nucleophilic substitution with silyl chloride and for Suzuki/Negishi type Pd-catalyzed cross-coupling chemistry through transmetalation to B or Zn species as well as direct coupling with aryl chlorides, while the reaction conditions used for the cross coupling were almost optimized in the authors' previous works (Nat Catal 2019) and substantially the same protocols were adopted in these parts.

The key enabling the advance of the present work is use of neopentylsodium instead of conventionally used alkyl halides such as pentyl chloride, to avoid undesired side reactions and decomposition through beta-H elimination. I consider that the methodology presented here should be certainly useful as a new protocol for generation and use of neopentylsodium for halogen-sodium exchange. I thus recommend this work for publication in *Communications Chemistry* provided that the following point on scholarly presentation is addressed in the revision process.

We thank the reviewer for the positive evaluation. We hope that the revised manuscript is suitable for publication in *Communications Chemistry*.

In the introductory part, the authors criticized Gilman's pioneering work for aryl-sodium exchange, which was reported in 80 years ago. Indeed the reported yield of 1-naphthylsodium was lower (28% yield), while the butylsodium was prepared from dibutylmercury and metallic sodium. I considered that the problem of the Gilman's protocol may be on preparation of butylsodium from dibutylmercury, but not on the halogen-sodium exchange based on the fact that the authors examined use of normal-alkyl sodium for halogen-sodium exchange (Figure 2, entries 1 and 2), which gave 2a in reasonable yields along with over-substitution product 3 formed to some extent. As the Gilman's report did not clarify the details on the side products, I would urge the authors to revise this part by removing the authors' speculation.

We thank the reviewer for the comment. Problematic side reactions during halogen-sodium exchange (such as alkylation) are also noted in other references (ref. 14 and 16), and observed by us (Fig. 2, entries 1 and 2). Therefore, while speculative, we believe that

a discussion on the possible reasons for the low yield is informative for the readers, and we kept the original sentence, while mentioning its speculative nature; we also added a comment on the different method Gilman used for preparing butylsodium. We removed the speculative statements from Fig. 1b.

Response to Reviewer 2 comments:

Comments:

This is an extension of the authors previous work on arylsodium compounds that reported their preparation by reducing aryl chlorides with fine sodium dispersions and their application in Negishi, Suzuki–Miyaura, and cross-coupling reactions. The new improvement here is simply to use neopentylsodium to generate the arylsodiums in situ, and although this involves an extra step, it opens the door to a wider range of arylsodium compounds for onward reactivity via the same Pd catalytic and cross-coupling methods used previously. The scope of the halogen–sodium exchange methodology between aryl and alkenyl halides and neopentylsodium covered in figure 3 is impressive and is probably the best selling point of the work. This is original new chemistry which advances the use of organosodium compounds in situ in organic synthesis so merits publication.

We thank the reviewer for the positive evaluation. We hope that the revised manuscript is suitable for publication in *Communications Chemistry*.

(A) Specific Points on this paper:

(i) The paper states “organosodium reagents have limited success in synthetic organic chemistry”. This requires elaboration to those chemists unfamiliar with the area. For example it should be explained that they have significant solubility problems; greater ionicity than RLi so generally are over reactive; and that ^{23}Na NMR is generally not very informative compared to the wealth of information available from ^7Li and ^6Li NMR studies, which makes mechanistic interrogation of organosodium reactions much more challenging.

We thank the reviewer for the comment. As kindly suggested, we have added a discussion on the drawbacks of organosodium compounds to the introduction.

(ii) “The direct use of organosodium compounds in cross-coupling reactions circumvents the need for additional transmetalation”. Progress in this objective would add to the interest and sustainability theme of the work. Unfortunately, this is limited to a single specialized case involving phenanthrenylsodium. Readers are left pondering, “what happens with the other sodium aryls mentioned in this paper?” Have they been tried?

We have added four new examples for the direct cross-coupling of organosodiums to Fig 7. They include 1-pyrenylsodium (i.e., **6b**), 4-*tert*-butylphenylsodium (**6c**), 2-trifluoromethylphenylsodium (**6d**), and (5-methylthiophen-2-yl)sodium (**6e**).

(iii) On reflection if this was my work I would publish it as a full paper rather than a short communication. There are important aspects of the work which are relegated to the supporting information that should be more prominently displayed in the paper proper if this area is to be taken up by the organic synthetic community.

For example:

“Caution: organosodium compounds are insoluble in hexane, therefore efficient and vigorous stirring is very important”.

“The quality of sodium dispersion gradually deteriorates upon contact with air.”

“Once exposed to air, it is recommended that sodium dispersion should be used within two weeks”.

“Neopentylsodium was stable at least for 1 h at 0 °C in hexane”.

We thank the reviewer for the comment. We have added information regarding the handling and safety of sodium dispersion to the manuscript. We have also added several photos to help readers visualize the procedures to the supplementary information (Figures S2, S3, and S4).

(iv) The chlorosilane used in the electrophilic quenches is always Me₂PhSiCl. What happens with Me₃SiCl?

We used Me₂PhSiCl as the silicon electrophile because it was our first choice following our previous work (ref 17, *Nat. Catal.* **2019**, 297), and it reacted almost quantitatively. We examined the reactions of phenanthrenylsodium with other electrophiles such as Me₃SiCl (**2c'**) and dodecylMe₂SiCl (**2c''**) to find that the silylated products were obtained in 82% and 85% yield, respectively. We have added these results to the manuscript.

(v) The successful multi-sodiations of arenes is another impressive aspect of the study, which extends the potential impact of the work to material science. Other important strategies involving sodium could be cited here such as Martinez's work in *Science Advances* 2017, Vol. 3, no. 6, e1700832 “Multisodiation: Templated deprotonative metalation of polyaryl systems: Facile access to simple, previously inaccessible multi-iodoarenes”.

We have added the suggested work as reference 31, and the related study as reference 30.

(B) General points on influencing the thinking and take up by the field.

These are also important considerations of practicality that will influence whether organic chemists will embrace the use of such organosodium methods. How can the sodium dispersion be made in the laboratory? I cannot find these details in the manuscript or supporting information except that it is available from some commercial suppliers. This is critically important as the particle size of the sodium dispersion must be smaller than 10 μm and its concentration in paraffin must be controlled or else it could become very hazardous in terms of potential fires. The authors state “It is non-pyrophoric and stable under air. Because of the lower reactivity of the sodium dispersion at this concentration on contacting water, in Japan, the sodium dispersion is categorized as a less hazardous material than commercial butyllithium”.

We used sodium dispersion produced by KOBELCO ECO-Solutions Co. It is a high quality sodium dispersion in paraffin oil and has a particle size smaller than 10 μm . The same sodium dispersion is now commercially available from Tokyo Chemical Industry Co., Ltd. [code: D5792] and FUJIFILM Wako Chemicals [code: 638-46321] all over the world.

We believe that one of the key reasons why organolithium compounds have become indispensable tools in (in)organic synthesis is because various organolithium reagents including BuLi are commercially available. We hope that the preparation of organosodiums using commercially available sodium dispersion will become a routine and reliable task in the future, as it is the case with the well-established preparation of organolithiums.

We have not prepared sodium dispersion by ourselves, but the preparation of sodium dispersion or fine sodium sand has been known since 1937 according to ref. 11. It is typically prepared by vigorously stirring molten sodium in a hydrocarbon solvent at temperatures above the melting point of sodium (ca. 98 °C). Some examples: Morris, M. J. *et al. Inorg Chem.* **45**, 10967 (2006); Mioskowski, C. *et al. Angew. Chem. Int. Ed.* **41**, 340 (2002); Marguerite, J. *et al. Org. Synth.* **63**, 147 (1985); Nelke, J. M. *et al. Org. Synth.* **57**, 1 (1977). The preparation of alkylsodiums with sodium dispersion is also introduced in ref 2 (Schlosser, M. *Organometallics in Synthesis: A Manual* 2nd edn).

(i) For balance, it should be pointed out that organosodium compounds are generally more pyrophoric than organolithium compounds, and while the authors are discussing the sodium dispersion here they do not mention the potential pyrophoricity of the neopentylsodium reagent nor that of the alkenyl- and aryl-sodium intermediates they are producing following the initial dispersion step of the reaction. These comparisons should also be included in the paper.

Please note that we directly handle only sodium dispersion, and the generated organosodium reagents are kept under an inert atmosphere within the reaction vessel until the termination of the reaction, minimizing hazards.

But we do agree with the reviewer's comment on the generally higher reactivity of organosodium, and recognize its potential pyrophoricity especially at high concentrations.

In order to minimize these potential risks, we keep the concentration of organosodiums low. Organosodium compounds (neopentyl-, aryl-, alkenylsodiums) generated in situ using SD (non-pyrophoric under air) have typically concentrations of 0.2–0.4M throughout this study. At this concentration, the reactivity of organosodium is quite mild (and milder than commercially available 1.6M BuLi solution). We have modified the comments regarding the safety and reactivity of organosodiums in Ref 19.

(ii) Figure 1b states “RLi highly pyrophoric”. Again for balance, should neopentylNa and the RNa not also carry such warnings?

As discussed above, the concentration of organosodiums was kept low (0.2–0.4 M) throughout this study.

Importantly, the organosodium intermediates were generated and reacted in-situ under an atmosphere of inert gas, and we did not transfer them using a syringe/cannula from another reaction vessel or a commercially available reagent bottle.

But we do agree with the reviewer that hazards should never be underestimated, and we have added a comment regarding the potential pyrophoric nature of organosodium compounds to ref 19.

In summary, while the results in this study are important since they are building towards the aspiration of the development of an organosodium “technology”, the paper as written at present requires modification to give a more complete picture of the story so far focusing on important limitations as well as the promising advantages of the different types of organosodium compound that are involved.

We thank the reviewer for the comment. We hope that the revised manuscript is suitable for publication in *Communications Chemistry*.

REVIEWERS' COMMENTS:

Reviewer #2 (Remarks to the Author):

The authors have taken considerable care in answering in full the issues raised by both reviewers. This paper significantly advances the field of organosodium chemistry so I am pleased to recommend its acceptance without further change.

Response to Reviewer 2 comments:

Comments:

The authors have taken considerable care in answering in full the issues raised by both reviewers. This paper significantly advances the field of organosodium chemistry so I am pleased to recommend its acceptance without further change.

We thank the reviewer for the positive evaluation. We hope that the revised manuscript is suitable for publication in *Communications Chemistry*.